# Using simulations to compare the current Davis Cup ranking system to Elo

**John Kelley**  *

The Advanced Wellbeing Research Centre, Sheffield Hallam University, Sheffield, United Kingdom

* j.kelley@shu.ac.uk

## Abstract

The Davis Cup is the premier men's team event in tennis, run by the International Tennis Federation and in which over 130 nations compete. It uses a merit-based ranking system that allows nations to gain a pre-determined number of points when they win. The rankings are integral to the competition structure and used in the draws of every round. Therefore, it is essential that the ranking method performs well with respect to required performance criteria of the International Tennis Federation. The Elo rating method is a commonly used method of rating and ranking participants in a competitive exercise and is used by FIFA for the ranking of male and female national teams. The performance of the current Davis Cup ranking method and Elo rating were compared using a simulation of the Davis Cup competition structure. Four criteria were used for the comparison: Finishing Order Correlation, Skill Level Correlation, Responsiveness, and Protection. Such a comparison has not previously been published. The two methods were comparable across three of the criteria, with the Davis cup easily outperforming Elo in responsiveness. Indeed, the Elo method had such poor responsiveness that improved performance may not be fully recognised within a player's career. An alternative method of optimising the Elo K-factor parameter was developed and this improved the performance of Elo to match the current Davis Cup method. In conclusion, the current Davis Cup ranking method is performing at a standard that cannot be matched by typically optimised Elo but can be matched when an alternative optimisation method is used. Therefore, no evidence was found to suggest that the current Davis Cup ranking method could be improved upon by using Elo. However, alternative K-factor optimisation methods should be considered when applying Elo to a competition.

## 1 Introduction

A ranking system attempts to order participants in a competitive environment (team or individual) based on performance levels. It will exist as a tool for a sport or game because the raw finishing order in competitions doesn't represent the competitive field. This paper focuses on a competitive environment where single stand-alone ties are played by 2 participants at a time in a field of many participants.

If there is no rigid structure to the number of times participants play each other, then the results themselves cannot define a complete competitive order. Therefore, a method that

**Data Availability Statement:** The data and simulation code are stored in the Sheffield Hallam University Repository: https://shurda.shu.ac.uk/id/eprint/195.

**Funding:** Sheffield Hallam University financially paid for by the International Tennis Federation

provided support in the form of salaries for JK. The specific roles of this author is articulated in the 'author contributions' section.

**Competing interests:** The authors have declared that no competing interests exist.

calculates rankings is required. When using a well-defined round-robin format where participants compete against each other opponent the same number of times, a form of ranking is usually still required. Typically $X$ points for a win ($Y$ for a draw if possible) are awarded and rankings based on the total number of points. Another well-defined structure is a purely knockout competition that eventually crowns the 'best' participant, after winning a 'final'. In such competitions, using the competition results to rank the participants is often not useful as many teams have an equal result. The 1st and 2nd placed participants can be defined from the final, with the losing semi-finalists being equal 3rd, the losing quarterfinalists being equal 5th, etc. As you continue down the knockout levels, an increasing number of participants would be assigned ranks identical to others and the ranks fail to differentiate between participants effectively. When rankings are needed for such competitions, for example when drawing ties, a previous finishing order cannot be used and a method or system for determining each participant's rank is required. The Davis Cup is a complex combination of many levels of round-robin format and knockout format and the number of ties nations play can vary.

## 1.1 General expectations of a ranking method

An obvious and reasonable expectation is that the rankings will well-represent competitive results. If a finishing order of a competition can be defined, then the ranking method order should correlate highly with the finishing order of the competition. This is something that can be practically tested from real competition data. However, if competition data can be simulated it allows a larger dataset to be used.

Seeding in any competitive draw is used to well distribute nations so that the 'best' nations do not meet and knock each other out. Rankings are often used to define who the 'seeded' participants are, and that is the case with the Davis Cup. In this paper, the term *skill level* represents the quality of a nation in the Davis Cup and so the best nations will have a high skill level. Therefore, when seedings are based on rankings, rankings should reward nations that have a high skill level with a better rank. Quantifying skill level is practically impossible in real competition. This makes it not practical to directly measure the correlation between rank and skill level. In a computer simulation of a competition, the skill level of a participant can be defined and quantified and used to determine simulated results. Therefore, simulations would allow such correlations to be measured. Simulations-based optimisation has been previously used in medical supply chain design [1].

## 1.2 Description of the Davis Cup

The Davis Cup [2] is the premier international team competition in men's tennis and over 130 nations compete at various levels of the competition structure. It is run by the International Tennis Federation (ITF). The current competition structure has been in place since 2019 following a major change in format and is shown in Fig 1. Some small iterations have been made since the change in format with the most significant change being 16 finalists from the end of 2022 onwards.

The competition structure means that some nations may play two ties in a year (any *qualifying round* into *Group I* or *Group II*) and some nations may play up to 6 ties (*Finals Qualifying* into the *Finals* and playing in ever round of the *Finals*). Wildcards are picked typically from nations in the Finals Qualifying before the 'draw' that determines which nations play in each tie.

The draws for ties in each knockout section of the competition structure use seedings. Within a knockout section of the competition, nations are split into two equally sized groups using world ranking positions. The nations in a seeded group will all have a better ranking than those in the non-seeded group. Seeded nations are drawn to play against non-seeded

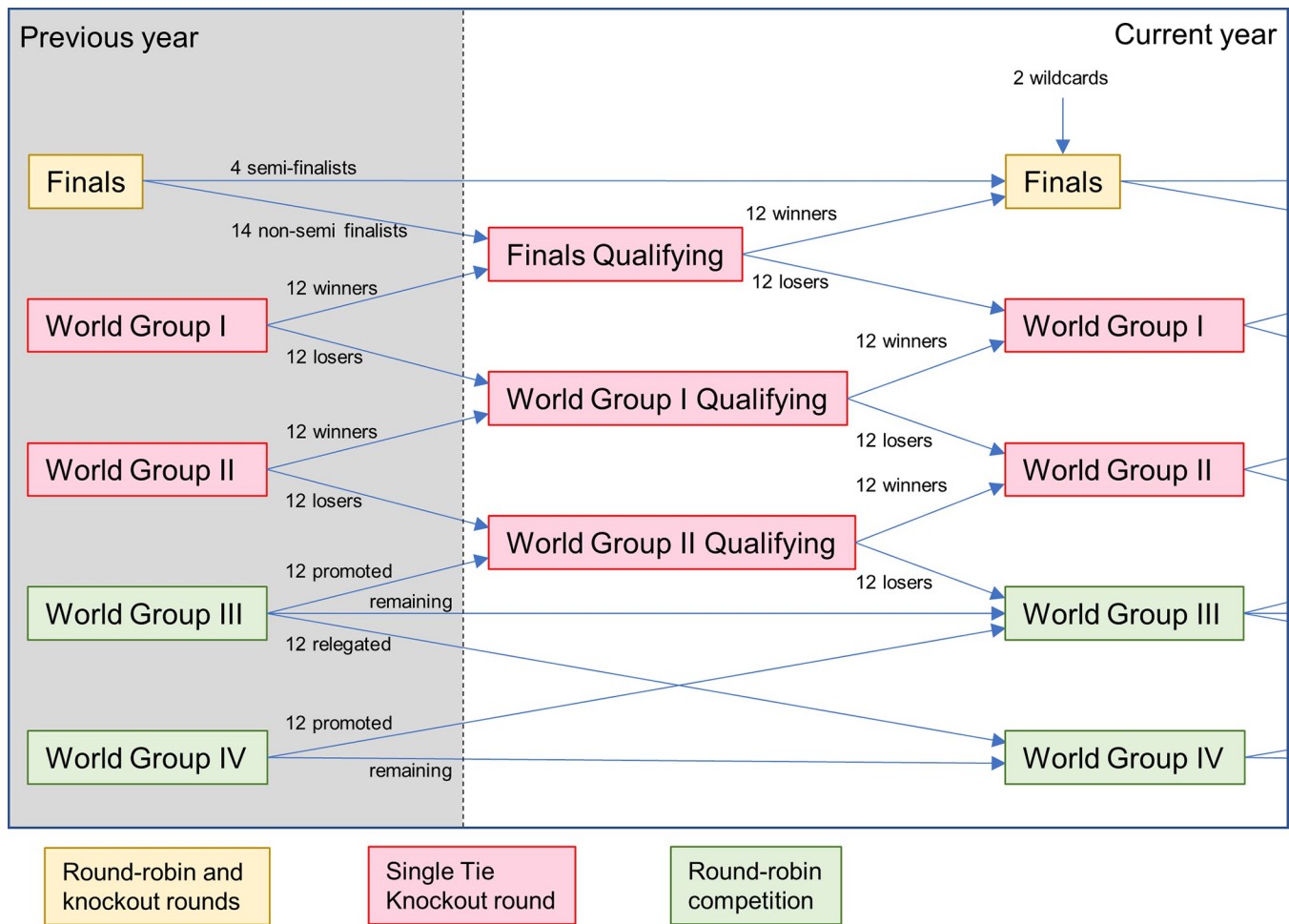

**Fig 1. The Davis Cup competition structure.** The flow of nations through the Davis Cup competition structure from the previous year and through the year.

nations. The *Finals* include a group stage in with 6 round-robin groups of 3 nations. From 2022, this has changed to 4 groups of 4 nations. These groups are also drawn based on three levels of seeding. This seeding is again determined by nation rankings. The round-robin groups in *World Group III* and *IV* are broken down by region and rankings are used to seed nations prior to pool draws.

Occasionally the ITF needs to make an 'adjustment' to account for a sub-optimal distribution of nations in the competition structure. This can happen for some of the wildcard choices, which is often used to guarantee the host nation is in the *Finals*, when they come from outside of the *Finals Qualifying* section. Also, nations may not be able to compete in the competition for a year, which would need an ITF adjustment. This can frequently happen in Group III and IV and nations in higher sections of the competition occasionally cannot compete. Adjustments move nations from one section of competition to another and the choice of nation is based on their rank.

### 1.3 Specific requirements for the Davis Cup ranking method

A nation can play a tie in the Davis Cup with as few as three players, although five players are typical. Therefore, it is possible that a nation's skill level could drastically improve if one or two

high-quality players start playing Davis Cup. This improvement is likely to last for the length of the career of those players. Other factors that could improve a nation's skill include increased investment that improves facilities and coaching available for the players. A ranking method should be responsive to such occurrences, with improved nations achieving an improved ranking within an appropriate response time. Given that a player's career in tennis could be reasonably expected to be 10 years, such a response should happen within half a career span, so 5 years.

For a nation without a large squad of highly ranked players to choose from, the effect of the nation's best player not playing and being replaced by a much inferior player would be large. It would cause a large drop in the nation's skill level and chances of winning ties. High-profile examples in recent years include Great Britain with Andy Murray and Switzerland with Roger Federer. Therefore, the pressure to compete is high and can conflict with wellbeing considerations such as sufficient recovery from injury and the option to take breaks from the game, for example paternity leave or to maintain mental health. To provide some relief from this pressure, the Davis Cup ranking method should afford some protection to a nation's ranking if there is a temporary drop in skill level. An injury may lead a player to miss most of the ties in a year and paternity leave could be up to a year. Therefore, one year is an appropriate time span for a temporary drop in nation skill level as a year.

The Association for Tennis Professionals (ATP) allow players to apply for singles or doubles ranking protection when they are injured [3] and the Women's Tennis Association extends a similar rule to maternity leave [4]. These protections freeze the individual's ranking until they return. For a Team event such as the Davis Cup, the nation ranking cannot be frozen in such situations because the nation will keep playing while certain players cannot play.

## 1.4 Ranking method options

This paper considers two ranking methods. The first is the ranking method currently used by the Davis Cup, which is a merit-based method. The second is the Elo rating system which is becoming more commonly used in sport.

**1.4.1 Merit-based ranking.** A merit-based ranking method awards a set number of ranking points for playing in a tournament and winning matches in a tournament. For example, the Women's Tennis Association's singles ranking method awards 10 points for playing (and losing) in the first round of a Grand Slam tournament, the highest level of competition in tennis. The losing finalist receives 1300 points, and the winning finalist receives 2000 points. Lower-level tournaments award fewer points but in a similar way. A player's highest scores from 16 tournaments over the latest 52-week period is included in their total ranking points.

The Davis Cup uses a merit-based ranking method for the competing nations similar to that used in singles rankings. However, with nations playing much fewer ties throughout a year than there are matches played by an individual, the points from each tie over a 4-year period are included with each previous year discounted by 25% (points from the past year worth 100%, points from the year before worth 75% etc.).

The ranking method includes bonus points for away wins because of the advantage that being a home nation in a tie conveys–the home nation determines the location and court surface to be played on and will ensure that the surface suits that nation's players. It also includes a bonus for beating highly ranked nations. The full description of how ranking points are calculated is presented on the Davis Cup website [5].

**1.4.2 Elo.** The Elo rating system, originally designed for chess [6], rates individuals from of a group of participants who interact by playing head-to-head ties. Each participant has a rating in the form of a single value and the difference in ratings between the two players is used to

predict the outcome of a match. The ratings of the two participants changes based on comparing the actual result to the predicted result.

FIFA, the international governing body of soccer, use an Elo rating to rank the male national teams [7], modified to include weighting levels reflecting the importance of the matches. FIFA use a similar weighted Elo rating system to rank female national soccer teams. The structure of international football competitions has some similarities to the Davis Cup. Both have qualifying stages to finals events (soccer World Cup/Davis Cup Finals) and there are round-robin groups and knockout ties at different stages of the general competition structure and at the finals event.

Elo is often used as alternative ranking methods not endorsed by governing bodies. For example, in basketball [8], for predicting playoff results and in American Football [9]. Elo rating has been used to predict future performance in singles tennis [10] which showed that Elo provided a better "measure of likely performance" than using the official rankings.

Because of its high-profile use by FIFA [7], its stability and statistical accuracy [13], and its performance in singles tennis [10], it should be considered as an alternative to the current Davis Cup ranking method.

An extension of the Elo system exists, called Glicko [11], which includes a confidence rating with the performance rating. Glicko is not considered here because of the added complexity of ranking participants using two rating values [12]. Also, the Davis Cup rankings can't exclude nations from a ranking list based on a confidence rating because the rankings are needed in the competition structure.

A potential limitation of Elo is that it is attempting to calculate a rating that represents the relative skill level of competitors based on prior results. In the Davis Cup, the competitors are the Nations and between ties in the Davis Cup, a nations actual skill level could change if different players are selected. This means the prior results may not well represent the current skill level of a nation and so the Elo rating would not well represent the current skill level. This is an issue in all applications of Elo. A chess player may be improving or declining in actual skill level and soccer team selections often change from match to match. However, the smaller team size in the Davis Cup may make this issue more pronounced.

This issue would be less pronounced if the rating was applied to individual players rather than nations. However, it is not feasible to combine playing ratings to form team rankings in this way. It would cause nation rankings to change when teams are selected rather than just when new results are achieved. Also, draws are made in advance of, and sometimes many weeks before, ties are played to ensure playing schedules and travel arrangements are set. Any team selection set at the time of a draw may not be valid by the start of a tie. Indeed, the result of the draw determines which nations choose the surface for home and away ties. Surface selection may heavily influence team selection as players can often be relatively stronger or weaker on different surfaces. Using individual player ratings may allow better prediction of results but would not provide a practical ranking system for Davis Cup nations.

## 1.5 Aim

This paper aims to compare the current Davis Cup ranking method to the Elo rating system using a computer simulation of the competition structure and results. To compare the methods, specific criteria are needed. Vaziri, B., *et al*. [12] stated properties that a ranking method should have–the method should i) consider the strength of the opponent, ii) provide an incentive to win and iii) consider the sequence of ties played by a participant. These criteria do not differentiate between our two methods as both give more points for beating a stronger opponent, both provide points for a winner (and Elo reduces points for the loser) and neither consider the sequence of ties played.

Xiong et al. [13] state that "a good balance between ranking accuracy and popularity promotion" is important. Popularity promotion is defined as increasing public interest in part of the competition. For example, a ranking method with more randomness might increase public interest in the draw process for the Soccer World cup. Popularity promotion will not be considered as a criterion for assessing outright performance of a ranking method since it is essentially a measure of how effective the ranking system might be as a promotional tool.

The previously described general expectations of a ranking method and specific requirements for the Davis Cup will be used as our criteria:

1. **Finishing Order Correlation**–Accurately reflect the annual competition finishing order

2. **Skill Level Correlation**–Accurately reflect a participant's skill level–this concept is not possible in reality but can be tested when competitions are modelled with results dependent on participant skill level.

3. **Responsiveness**–React quickly to a participant if their competitive standard changes–for example an improving team, through improved coaching, players or investment, should be able to improve their world ranking in a short time relative to their player careers.

4. **Protection**–Include short-term protection for participants–this is to reduce the pressure on athletes to compete every year, allowing for sufficient recovery times from injuries and allowing for breaks in play that may include maternity or paternity leave.

## 2 Methods

Historic results in the Davis Cup cannot be use to compare ranking methods because the competition structure relies on the rankings for seedings and draws. When using a different ranking method, the different seedings produced would lead to different draws and soon the historical ties would not well represent what should be played for the different ranking method. Additionally, the competition structure and current ranking method have been in place since 2019, with the previous competition structure and ranking method being radically different. The number of historic results is not sufficient to do a thorough comparison.

### 2.1 Modelling the Davis Cup

The competition structure shown in Fig 1 was modelled using MATLAB. To match the usual number of nations competing in the Davis Cup, 132 competing participants were appropriately distributed across the competition structure. Each participating Nation was given an initial rank position and a skill level value. The number 1 ranked nation was given a skill level of 150, the number 2 ranked nation was given 149 and so on. Initial draws for each stage of the competition were made based on these initial ranking positions.

The results of ties were randomly determined with a bias towards nations with higher skill levels. If two competing nations have equal skill levels, they both have a 50% chance of winning. As the difference in skill level increase, the percentage chance of the most skilful nation winning increases by 2% for each point difference up to a maximum chance of 90%. This means that a skill point gap of 20 or more points will give the nation with the higher skill level a 90% chance of winning the tie. This relationship between skill level and result probability was chosen because 20 skill points is approximately equivalent to 20 ranks or positions. The Finals contains 18 nations and Group I, Group I Qualifying, Group II and Group II Qualifying contain 24 nations. Therefore, 20 ranks or positions represents moving down the competition structure by roughly one full level. This means, the maximum likelihood of winning is reached when playing a nation that that should be about one level of the competition lower. The

maximum win likelihood was set at 90% to represent the chance of an upset that could happen when, for example, nations that expect an easy victory don't play their best players.

Maquiriain and Baglione [14] showed that there is an advantage to nations playing at home in the Davis Cup. To account for this, a 15% home advantage was added to the home nation's chance of winning (and subtracted from the away nations chances) for ties where there was a home and away nation. The maximum chance of winning was always 90%. The design of the win likelihood and the influence of the home advantage was guided by consultation with the ITF.

Winning and losing nations progress through the competition as shown in Fig 1. After the first year was completed, the initial rankings were optionally updated. This optional update allows the model to be run in three ways:

1. Test–Without an associated or linked ranking method. Nation rank positions are fixed to their starting value. This can be used to generate a semi-representative set of ties and results.

2. Fixed ranking–With an associated ranking method, calculating rankings each year but NOT updating the initial rankings, which continue to be used for all draws. This allows different ranking methods to be compared on identical ties as the same simulated results on the same draws can be used. However, this is not fully representative of the combined behaviour of a fully linked ranking method.

3. Full–With a fully linked ranking method, updating the rankings every year. This is fully representative of the behaviour of the competition structure when linked to a ranking method, but different ranking methods can be expected to order nations differently at some stage. From this stage onwards in the simulation, the draws will be different for the different ranking methods, and the results therefore must be different. This means ranking methods can't be compare on identical sets of simulated results when using a full update.

To optimise Elo parameters, several sets of results were created using 1 above. Both 2 and 3 above were used to compare the two ranking methods. Additionally, the simulation could be run with

A. fixed nation skill level or

B. with varying nation skill level.

With varying skill levels, the model allows nations to become better or worse each year–each nation's skill level can change by a value between -3 and 3. The maximum skill level value was set at 160 and the minimum set at 0. While not specifically validated, this optionally allows for a more realistic nation behaviour. Düring, Torregrossa and Wolfram [15] used "intrinsic strength" similarly to skill level here to investigate modelling Elo rating with learning effects. However, the intrinsic strength only ever increases as competitors "learn" from competing and the increase dependent on the result and the strength of the opponent.

The simulation types will be denoted by 1A, 1B, 2A, 2B, 3A and 3B.

## 2.2 Defining the Elo rating system

Elo rating algorithms require two parameters, the K-factor, the $S$ parameter. The expected result, $W_1$ for participant 1 and $W_2$ for participant 2 of a tie, is determined using the $S$ parameter and is given by

$$W_1 = \frac{1}{\left(10^{\frac{(R_2-R_1)}{S}} + 1\right)} \quad \text{and} \quad W_2 = \frac{1}{\left(10^{\frac{(R_1-R_2)}{S}} + 1\right)} \tag{1}$$

where $R_1$ is the rating of participant 1 and $R_2$ is the rating of participant 2. The sum of $W_1$ and $W_2$ is 1 and the maximum value of each is 1. The closer either is to 1, the more that participant is expected to win. The $S$ parameter scales the difference between the rating values, so using a higher $S$ value makes the expected results less sensitive to differences between ratings. The change in ratings, $\Delta R_1$ and $\Delta R_2$, given the actual result, $w_1$ for $w_2$, are scaled by the K-factor:

$$\Delta R_1 = k(w_1 - W_1) \quad \text{and} \quad \Delta R_2 = k(w_2 - W_2). \tag{2}$$

The K-factor scales how many points are gained or lost by multiplying by the difference between the actual and expected results.

Let's consider $R_1$, $R_2$ and $S$ in Eq 1; if we scale them all by the same factor, $F$ say, the result for $W_1$ or $W_2$ is unchanged. If we scale the ratings $R_1$ and $R_2$, the change in ratings $\Delta R_1$ and $\Delta R_2$ should be scaled by the same factor, so to do that, $K$ should be scaled by the same factor. Therefore, using a K-factor of $k$ and an $S$ parameter of $s$ will have identical behaviour to using $Fk$ and $Fs$ except the ratings and rating changes will also be scaled by $F$. Therefore, we can fix the $S$ parameter at 400, as is typically used, and optimise for the K-factor.

The standard method of determining the value for the K-factor is to choose the value so that the prediction (Eq 1) favours the winning participant as often as possible. All participant ratings are initially equal, and participants may need to play many times for their rating to be become representative of their ability relative to the competitive field. Results before the ratings are representative should not be included when determining the best K-factor. Therefore, a large set of representative results for a representative group of participants are required, enough so that each participants' rating becomes representative of their performance plus enough to determine the K-factor.

The competition model was used to produce an ideal set of results to optimise the K-factor. The simulation type 1 was run 100 times over 200 years to create a set of results. The Elo method was used to calculate nation ratings based on their results, with each nation's rating starting the first year at zero. The first 100 years of each run were ignored to allow the ratings of nations to become representative—nations will have played at least 200 ties to develop their rating. The predictions from the Elo method were compared to the actual simulated results for the remaining 100 years of each simulation. The K-factor was varied to find the highest percentage of correctly predicted tie results–ties won by the Nation with an Elo predicted result above 0.5. For optimisation, the cost function that needed to be minimised was the corresponding percentage of incorrectly predicted results.

As previously discussed, a nations actual skill level may vary between ties due to team selection. The simulation allows for a nations skill level to vary. There is a potential discrepancy between the skill levels of the nation during the previous ties, the results of which are used to calculate the Elo rating, and the current skill level of the nation. This discrepancy would cause the Elo rating to not best predict the outcome of the next tie which may mean the standard K-factor optimisation does not produce optimal performance. Additionally, the requirements of the Davis Cup rankings are not to best predict future results, but to perform well against the stated criteria. The standard Elo optimisation may not produce the optimal Elo K-factor for the stated criteria.

An innovative alternative method is to choose a K-factor value that gives the best performance given the criteria. It is not possible to quantify the Responsiveness and Protection criteria in such a way that can be practically used in a cost function to optimise for k. The *Finishing Order Correlation* and *Skill Level Correlation* criteria are ideal metrics to use in a cost function. The best performing K-factor value would have the highest (positive) correlation value which is the equivalent of minimising a cost function that uses the negative of the correlations.

Again, the k-factor was varied to minimise the cost function on the same set of 100 simulations.

Both optimisations were repeated 5 times to ensure that the behaviour of the cost function as the K-factor varies was consistent for different sets of simulated results.

## 2.3 Ranking method comparison

The Davis Cup ranking method and the two Elo methods using the K-factors from the two optimisation methods were compared across each of the four criteria. For the *Finishing Order Correlation* and *Skill Level Correlation* criteria, the simulation was run 100 times for 110 years for the 4 different simulation types 2A, 2B, 3A and 3C. The first 10 years of each run were ignored to allow the ratings of nations to become representative. Only 10 years are required here because the nations can be given an initial rating based on the distribution and spread of the ratings found during the optimisation process. For any given year of the simulation, the *Finishing Order Correlation* was determined by calculating the Spearman's rank correlation between the order of the rankings and the finishing order of the competition. The *Skill Level Correlation* was determined by calculating the Spearman's rank correlation between the order of the rankings and the order of the nation's skill level, starting with the highest skill level. The mean of both correlation values from every run and year was calculated.

The *Responsiveness* of the rank position of a nation that has its skill level increased by 40 points was considered. Given that the initial skill level range was 1 per rank and the maximum and minimum skill level values had a range of 160 for 132 nations, 1 skill point was approximately 1–1.2 ranking positions. The expectation was that a 40-point skill increase would expect a 30–40 rank position improvement. A simulation was run for 60 years and then split into 2 simulations, one where a selected nation has 40 points added to its skill level and one with no adjustments–the control. Both simulations are then continued and the number of years it takes for the adjusted nation to improve by 20 ranking places compared to the unadjusted simulation was recorded. Twenty ranking places was chosen as it was a significant proportion of the expected ranking improvement of between 30 and 40 places. This was repeated 100 times for a nation and the average number of years to improve by 20 places calculated. Ten nations were selected, spread across the rankings. To perform well, the rank position of those nations should respond quickly to the increase in skill level, improving to represent the new skill level.

For the *Protection* criteria, the number of years it took for a nation's ranking to recover after a temporary drop in skill level was measured. The skill level was reduced by 20 points for a single year before increasing by 20 points to return to the same level. The simulation was run in a similar way to the method used for *Responsiveness*. In this case, the rank in the simulation was compared to the rank in a parallel control simulation without the skill level adjustment and the number of years until the rank returns to within 2 places of the control simulation was recorded. Again, the simulation was run 100 times for each selected nation and 10 nations were selected, ranked 1, 11, 21, . . .., 101.

## 3 Results

### 3.1 Elo optimisation

**Standard optimisation.** The general behaviour of the standard cost function as the K-factor value was varied is shown in Fig 2. The behaviour was very similar for each set of simulated results with the same simple general shape and the same region of high-performing K-factor values. The cost value did not vary significantly for K-factor values close to 28. This lack of variation for a range of K-factor values means that any value in that range had similar performance. There was a tendency for local minimums that became apparent when higher

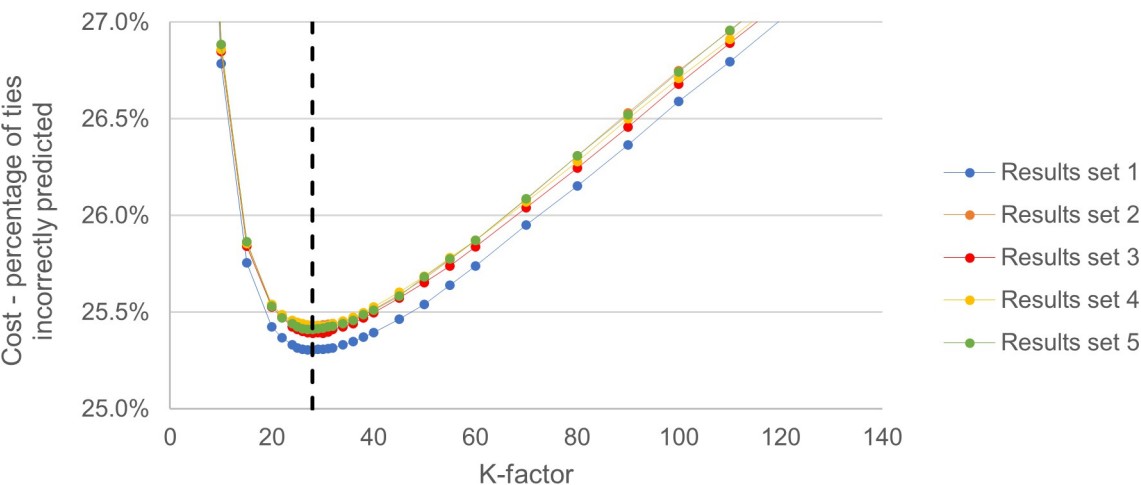

**Fig 2. Standard optimisation for Elo K-factor.** The cost defined as the percentage of incorrectly predicted ties, at specific K-factor values for each set of results. The final chosen K-factor of 28 is shown by the dashed line.

resolutions of K-factor values were used. This meant that optimisation algorithms gave multiple different results in the range of between 25 and 31, depending on different initial conditions, all with essentially the same cost value. Typically, an integer value for k is used, so 28 was chosen.

**Alternative optimisation.** The general behaviour of the alternative cost function as the K-factor value was varied is shown in Fig 3. The behaviour was again very similar for each set of simulated results and has a simple general shape. The cost value does not vary significantly for K-factor values in the region of 125–130. As in the previous optimisation, an integer value was chosen, 128 in this case.

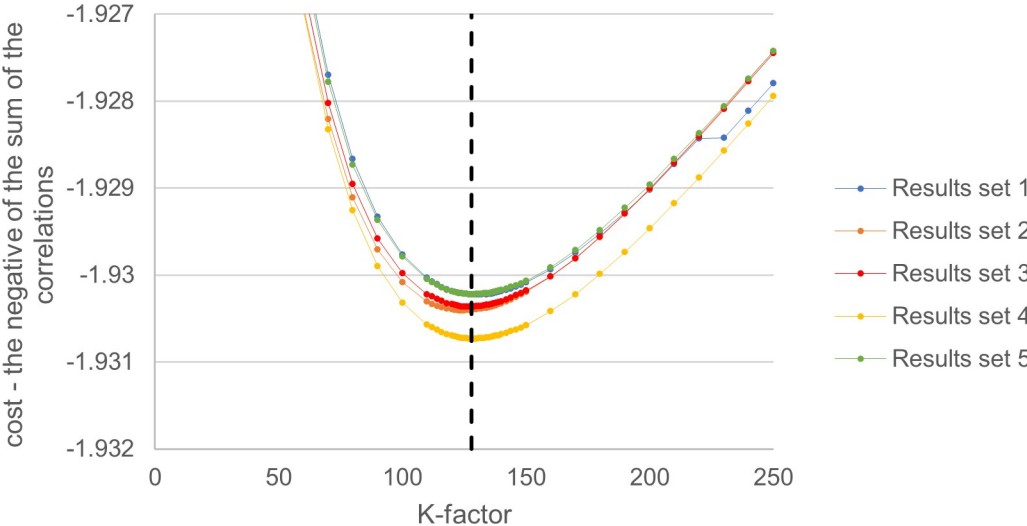

**Fig 3. Alternative optimisation for Elo K-factor.** The cost at specific K-factor values for each set of results. The cost was defined as the mean sum of the two negative correlations: 1. Between the order of the rankings and the finishing order of the competition; 2. Between the order of the rankings and the order of the nation's skill level for each year. The final chosen K-factor of 128 is shown by the dashed line.

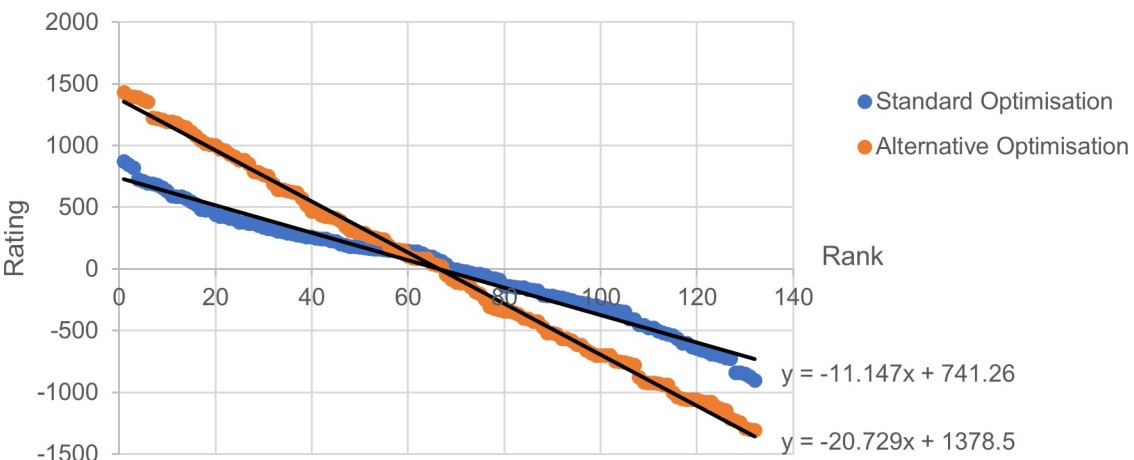

**Fig 4. Elo rating against rank.** The rating value plotted against ranking for all nations at the end of a 200-year simulation.

## 3.2 Rating distributions

After a 200-year run of the simulation, the rating distributions for both Elo rating methods are shown in Fig 4. The distribution is close to linear for both, and the gradient and intercept values of the trend lines can be used to give an appropriate initial rating for each nation. A real, practical implementation of Elo needs initial values such as these. Setting initial rating values based on the behaviour shown in Fig 4 allows simulations to be representative much more quickly, drastically reducing the number of years initially ignored. Initial ratings were set to match these trend lines for all subsequent simulations.

## 3.3 Finishing order correlation and skill level correlation criteria

Tables 1 and 2 show that there was very little difference in performance between the three methods and that all the mean correlations are high. No single method was consistently the worst across both criteria.

## 3.4 Responsiveness criterion

Fig 5 shows that the standard Elo rating method was much less responsive to an improved nation than the Davis Cup ranking method and the alternative Elo rating method. It took 2–3 times longer for an improved nation to gain 20 ranking places compared to when using the other methods. This was the case across all the selected nations and was more pronounced for nations with initial ranks and skill levels nearer the middle of the range. The alternative Elo rating method and the Davis Cup ranking method were evenly matched. Fig 5A) shows results

**Table 1. Finishing position correlations.** The mean correlation between end of year nation rank and nation finishing position across 100 simulations of 100 years. All the mean correlation are high with the alternative Elo rating method having higher correlations for each simulation type and the standard Elo rating method having lower correlations.

|  | *Davis Cup* | *Standard Elo* | *Alternative Elo* |
|---|---|---|---|
| *2A –fixed ranking, fixed skill* | 0.9586 | 0.9466 | 0.9607 |
| *3A –Full, fixed skill* | 0.9575 | 0.9469 | 0.9622 |
| *2B –fixed ranking, varying skill* | 0.9530 | 0.9381 | 0.9567 |
| *3B –Full, varying skill* | 0.9567 | 0.9438 | 0.9617 |

**Table 2. Skill level correlations.** The mean correlation between end of year nation rank and nation skill level rank across 100 simulations of 100 years. All the mean correlation are high with the standard Elo rating method having higher correlations for each simulation type and the Davis Cup method having lower correlations.

|  | *Davis Cup* | *Standard Elo* | *Alternative Elo* |
|---|---|---|---|
| *2A –fixed ranking, fixed skill* | 0.9772 | 0.9936 | 0.9802 |
| *3A –Full, fixed skill* | 0.9726 | 0.9934 | 0.9781 |
| *2B –fixed ranking, varying skill* | 0.9662 | 0.9863 | 0.9764 |
| *3B –Full, varying skill* | 0.9682 | 0.9860 | 0.9751 |

for when nation rank positions were fixed which allowed the tie results to be identical for the control simulation and for the main simulation until the increase in skill level occurred. Fig 5B) shows results for when rank positions were updated each year based on the ranking method. This is a more representative simulation but does not allow the same ties and results to be used for each ranking method. It shows similar behaviour to that seen in Fig 5A).

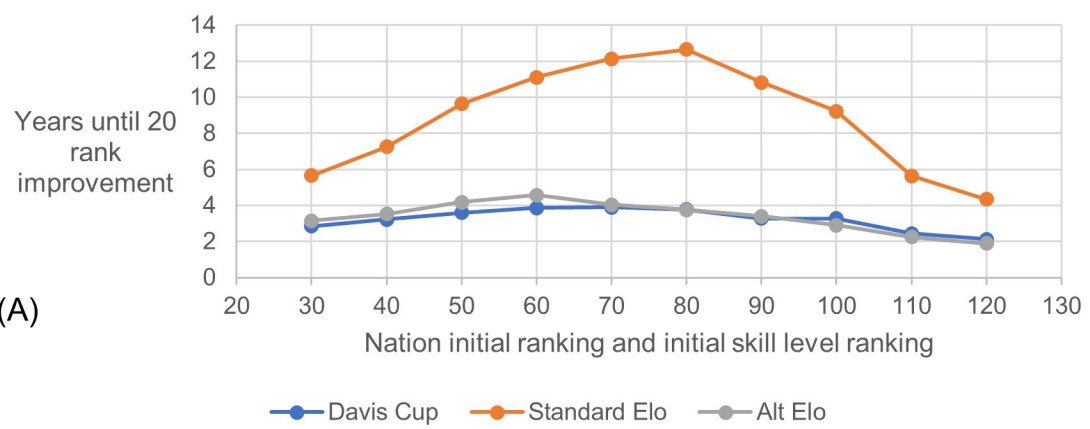

(A)

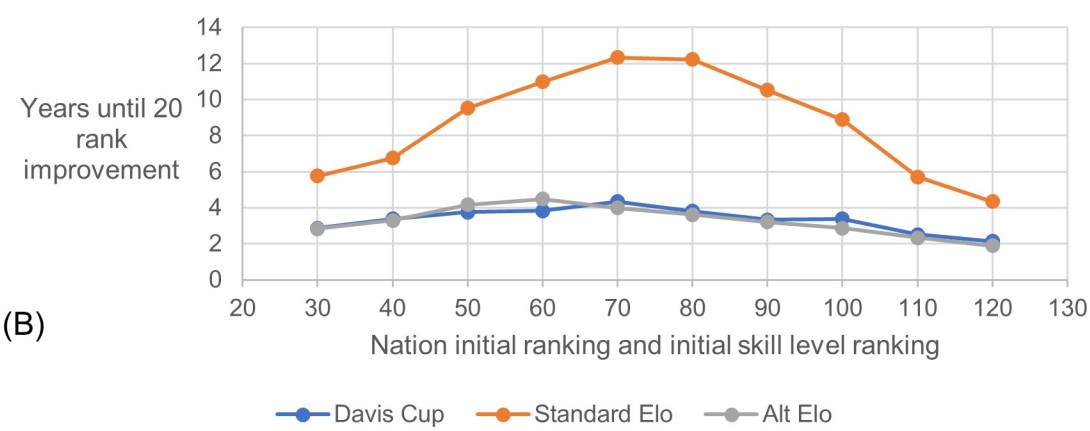

(B)

**Fig 5. Rank responsiveness to skill improvement.** The average from 100 simulations of the number of years for a nation's rank position to improve by 20 places after its skill level is permanently increased by 40 points. Ten nations were selected across the range of initial rankings (and skill levels) at 10-point intervals. (A) Simulation type 2A –fixed nation skill level and fixed nation rank position. (B) Simulation type 3A - fixed nation skill level and updating rank position.

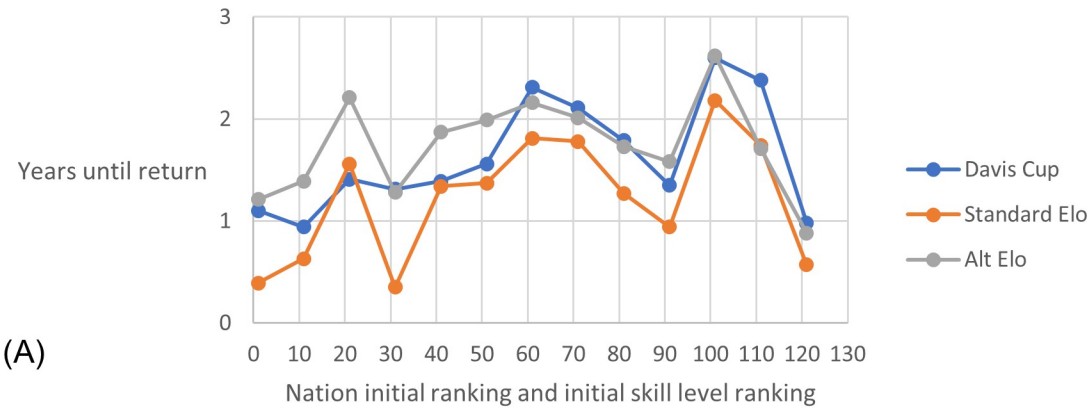

(A)

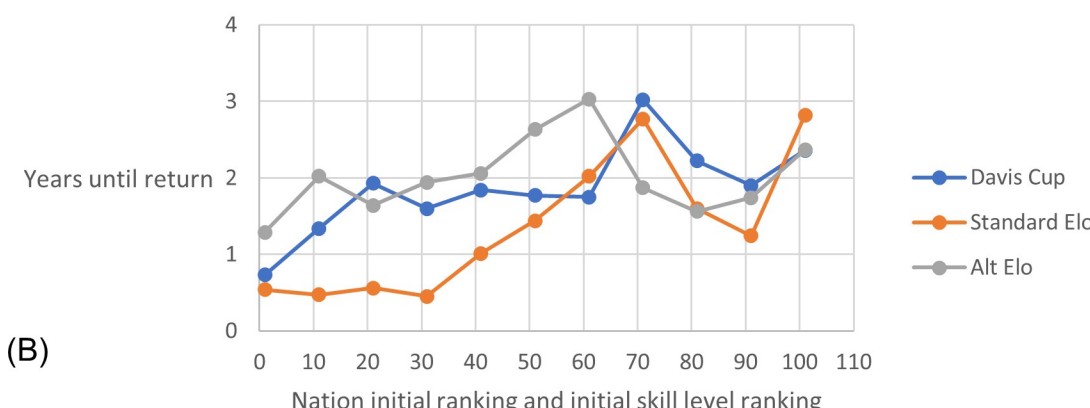

(B)

**Fig 6. Rank return time after skill level drop.** The average from 100 simulations of the number of years for a nation to return to within 2 rank positions of where they would have been ranked without a one year drop in skill level of 20 points. Eleven nations across the range of initial rankings (and skill levels) were selected at 10-point intervals. (A) Simulation type 2A –fixed nation skill level and fixed nation rank position. (B) Simulation type 3A - fixed nation skill level and updating rank position.

### 3.5 Protection criterion

Fig 6 shows that using all three methods provides similar levels of protection for a nation that has a temporary reduction in skill level. This was the case for both simulation types, with the number of years until the nation returns to within 2 ranks of its control ranking being three years or less across all selected nations. The standard Elo method appears to provide slightly greater protection than the other ranking methods.

## 4 Discussion

### 4.1 Elo optimisation

The two different Elo K-factor optimisation methods produced quite different K-factor values. The alternative method produced a K-factor value that was more than four times that of the standard method. This means a winning nation receives more than four times as many ratings points. The typical spread of nation Elo ratings, as shown in Fig 4, was also more spread for the alternative Elo method, although the gradient of the fitted line was less than twice as steep. Therefore, the alternative method causes nations to move up and down the ranking list more quickly, since the number of rating points they win or lose is proportionally higher than the

typical rating gap between nation ranks. More simply put, it would be more responsive to the result of each tie.

## 4.2 Finishing order correlation and skill level correlation criteria

There were only slight differences between the methods with all of them showing strong correlations between end of year rank and both the finishing position and skill level. Despite being optimised using these values, the alternative Elo method was not clearly better than the other two methods on these criteria, although it could be argued that it had slightly the stronger performance when considering both criteria together.

## 4.3 Responsiveness criteria

The standard Elo method was not responsive enough to an improved nation, with the average number of years taken for the nations ranking to improve 20 places after a 40-point skill increase being between 6 and 13 years. This would be a major part, if not all, of a player's career, which is too long. We can see that this is not caused by the competition structure itself since the other two methods allowed the nation's rank to improve rapidly, within 4.5 years for every tested nation. The difference in the responsiveness of the two Elo methods agrees with the expectations discussed in 5.1.

The standard Elo method was particularly unresponsive for nations with initial ratings and skill levels in the middle of the range. These nations are likely to be playing in World Group II and World Group III. After improving, they will likely be playing ties in the red region of competition (Fig 1) which means they would often compete in just two ties per year. This provides only two opportunities to improve their rating in a year. The low K-factor of the standard Elo combined with the limited number of ties played in the red region of the competition means that a nation cannot quickly improve their rating even if they win every tie.

The alternative Elo method matched the performance of the Davis Cup method despite not being specifically optimised for this criterion. This shows that the chosen optimisation criteria did not cause specifically poor performance in this criterion.

## 4.4 Protection criteria

The standard Elo method generally provided marginally more protection than the alternative Elo methods. This is because of the already identified reduced responsiveness. It is particularly protective for better ranked/higher skill level nations. During a 1-year drop in skill, these nations are likely to be playing in the red region of the competition (Fig 1) and so not able to play and lose many ties, so not lose many rating points.

The alternative Elo method again had similar performance of the Davis Cup method.

## 5 Conclusion

Neither Elo rating method produced a meaningful improvement over the current Davis Cup ranking method. The standard Elo method was shown to be unusable because of the lack of responsiveness to improved nations. The alternative Elo method performed at very similar level to the current Davis Cup method across all the criteria and was clearly better than the standard Elo method based on the *Responsiveness* criterion. This shows that it is possible that the Elo method can be tailored to requirements beyond just the best prediction of results. When defining Elo rating for other applications, a non-standard cost function choices for optimising the K-factor value should be considered. FIFA state that the method was chosen "After a long period testing and analysing the best way to calculate the FIFA/Coca-Cola World

Ranking" [16]. It is likely that such a thorough process included criteria such as those stated here. The K-factor is different for different levels of the competition but ranges between 5 and 60. Compered to our Elo methods, this is close to our standard method, so relatively less responsive than it could be. The FIFA criteria are likely to be more focussed on representing long-term performance and less focussed on short-term responsiveness.

There are potential avenues to improve the Elo method by modifying the weighting of results at different levels of the competition or adding in a points bonus for away ties. Considering the complex nature of the competition structure, choosing well-balanced and correct weightings is not trivial. The FIFA world rankings use weightings that effectively mean that the K-factor ranges from 5 to 60 depending on the importance of the match. The previous quote from FIFA [16] suggests that significant effort was taken to define these.

### 5.1 Limitations

In the Davis Cup, the away bonus would only apply to the red regions of the competition structure since these are the only home/away ties in the competition. Including an away bonus would likely increase the responsiveness of Elo for nations playing in the red region but also reduce the protection for nations in that region. Since it would potentially improve against one of our criteria and do the opposite against another, and since it is only relevant in a minority of our ties, it is not likely to improve performance.

The conclusions assume that the behaviour of the system would not change if different relationships between the win-likelihood values and nation skill levels were used and if a different distribution of skill levels were used across the nations. The skill level and related win-likelihood was defined with advice from the International Tennis Federation but has not been validated. When sufficient real competition data has been created, the skill level and win-likelihood calculation could be validated.

### 5.2 Practical applications and future prospects

The successful alternative Elo optimisation shows that Elo can be tailored for ranking method requirements rather than purely optimised for result prediction. Already implemented ranking methods based on Elo could be improved by considering such alternative optimisation. Ranking methods that are not based on Elo and that do not focus on predicting future results, may be improved upon by using an alternative Elo method.

The methods presented in this paper could be directly applied to the Billie Jean King Cup, an international team competition for women similar to the Davis Cup. The Billie Jean King Cup also recently changed competition structure. This means that there is a similar lack of real representative results. The Billie Jean King Cup uses a merit-based ranking method which a modified Elo method may be able to out-perform it against similar criteria used in this paper.

### Acknowledgments

The author acknowledges the International Tennis Federation for their contributions that ensured an accurate simulation of the Davis Cup was produced and that the criteria for ranking method performance were relevant.

## Author Contributions

**Conceptualization:** John Kelley.

**Data curation:** John Kelley.

**Formal analysis:** John Kelley.

**Investigation:** John Kelley.

**Methodology:** John Kelley.

**Project administration:** John Kelley.

**Resources:** John Kelley.

**Software:** John Kelley.

**Supervision:** John Kelley.

**Validation:** John Kelley.

**Visualization:** John Kelley.

**Writing – original draft:** John Kelley.

**Writing – review & editing:** John Kelley.

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
