## [Decision Letter · Decision Letter 0]

12 Jan 2023

PONE-D-22-29782

Using simulations to compare the current Davis Cup ranking system to Elo

PLOS ONE

Dear Dr. Kelley,

Thank you for submitting your manuscript to PLOS ONE. After careful consideration, we feel that it has merit but does not fully meet PLOS ONE’s publication criteria as it currently stands. Therefore, we invite you to submit a revised version of the manuscript that addresses the points raised during the review process.

Your manuscript has been assessed by two expert reviewers, whose comments are appended below. Reviewer 2 in particular has highlighted concerns about several aspects of the study design and how the reported research adds to the body of academic knowledge. Please ensure you respond to each point carefully in your response to reviewers document, and modify your manuscript accordingly.

We look forward to receiving your revised manuscript.

Kind regards,

Dr Joseph Donlan

Senior Editor

PLOS ONE

Journal Requirements:

https://journals.plos.org/plosone/s/file?id=ba62/dPLOSOne_formatting_sample_title_authors_affiliations.pdf

2. Please report the paid consultancy project information in the competing interest statements.

“The author(s) received no specific funding for this work.

The link to a paid consultancy project is described in the cover letter and repeated here: this research started during a period of paid consultancy by Sheffield Hallam University for the International Tennis Federation when designing the new Davis Cup competition structure and ranking method. The approach presented in this manuscript was used to provide evidence to the Davis Cup Committee prior to their choice of ranking method. The simulations and analysis have since been developed, including the concept of the alternative optimisation method to improve the performance of Elo rating.”

5. Please ensure that you refer to Figure 3 in your text as, if accepted, production will need this reference to link the reader to the figure.

Reviewers' comments:

Reviewer's Responses to Questions

**Comments to the Author**

1. Is the manuscript technically sound, and do the data support the conclusions?

Reviewer #1: Yes

Reviewer #2: Partly

2. Has the statistical analysis been performed appropriately and rigorously? 

Reviewer #1: Yes

Reviewer #2: No

3. Have the authors made all data underlying the findings in their manuscript fully available?

Reviewer #1: Yes

Reviewer #2: Yes

4. Is the manuscript presented in an intelligible fashion and written in standard English?

Reviewer #1: Yes

Reviewer #2: Yes

5. Review Comments to the Author

Reviewer #1: The reviewer would like to thank the journal for the opportunity to review this article.

The reviewer would also like to congratulate the author for having chosen as the subject of his study a considerably relevant aspect within the discipline of tennis, namely the analysis of ranking systems.

In fact, the issue of ranking systems for both individuals and teams participating in a competition is a very topical issue in the world of tennis due to the emergence of the ITF World Tennis Number.

Thus, the reviewer considers that we are faced with a tremendously current, modern and necessary study due to its obvious implications both for the participating federations and for the players who are members of the national teams.

The author adequately justifies the importance of the topic, explains its necessity and introduces it in an appropriate manner in the opening section of the article. Possibly the only suggestion of the reviewer would be the need to include enough references to support all the claims made in the first pages of the introduction, as the first reference is not included until well into the article.

In this sense, and in general, the reviewer considers that the total number of references is relatively small, so it would be advisable to add current and appropriate references that would provide solidity to the claims made in the document.

The research design, the hypotheses proposed, the sample, the analyses used, the presentation of the results and their discussion are appropriate for an article of this quality. It seems obvious that the author has mastered the subject, deals with it correctly and presents it appropriately so that it can be understood and followed by the reader.

The tables and figures used are clear, visually attractive and provide relevant information for the understanding of the manuscript.

As per the discussion, the author compares his results with those of relevant studies and carries out a very interesting exercise of reflection on his work.

As for the conclusion, the reviewer would suggest that the author dedicate a few paragraphs to three aspects that he considers fundamental in any research work. On the one hand, reference should be made to the limitations of the study. That is, those aspects that the author identifies as obstacles to the study and the reasons for them. On the other hand, a reference to the future prospects of the research, based on the results obtained in the study. Specifically, the directions or lines of work that could be derived from the findings of this research. And finally, it would be highly advisable to dedicate a few paragraphs to the possible practical applications of the work. A kind of answer to the question "What then? In this way, the manuscript could be brought to a more appropriate conclusion.

For all the above reasons, the reviewer considers that the submitted article can be accepted for publication in the journal.

Finally, the reviewer would like to thank again the author and the editor for their interest in publishing research articles related to tennis.

Miguel Crespo, PhD.

Reviewer #2: The authors want to determine whether an Elo-based approach for rating Davis Cup teams might offer an improvement over the Davis Cup ranking method. They use event simulations to evaluate the quality of ratings and rankings according to several properties they argue are desirable for a rating/ranking system. The study aim has potential interest for the implementation of the Davis Cup implementation. The authors don't provide a good case for the scientific importance of the study.

1. The major concern I have with the study is that the Elo rating is derived at the team level with no consideration of the skill of the players who compose the team. The power of the Elo system lies in its dynamic self-correcting update, where ratings go up or down based on the latest results in a direction to make the last prediction more correct. Because the Davis Cup team competition can change drastically from round to round, this raises the question of whether a team-level Elo is an appropriate application of the tool. The goal of the Elo system is to measure an underlying skill that can't be directly observed. What is the skill that is attempting to be measured here if the composition of the Davis Cup team is changing so frequently? The author would be on firmer ground if the simulation were based on the Elo-predicted results on individual players or, perhaps, some team-level rating based on the summary of the player skills on the team at any given time.

2. Rankings don't provide a direct way of simulating match results. The author uses skill points for each ranking to derive probabilities of team A beating team B but these are arbitrary. If other probabilities were assigned to the skill points, the results could change, which puts doubt on the validity and robustness of the findings.

3. Some of the criteria for a 'good' ranking system appear to be in conflict. Both 'Responsiveness' and 'Protection' could be in conflict with the skill correlation of the system, for example, if the system is too sensitive to small sample sizes in how it updates or provides a disincentive for the best players to participate.

4. A smaller point is that the scaling factor of the Elo system should also be considered a tuning parameter as this should approximate the variation in the underlying skill distribution.

6. PLOS authors have the option to publish the peer review history of their article (what does this mean?). If published, this will include your full peer review and any attached files.

Reviewer #1: **Yes: **Miguel Crespo

Reviewer #2: No

---

## [Author Response · Author response to Decision Letter 0]

21 Aug 2023

Response to reviewers attached as a word file.

---

## [Decision Letter · Decision Letter 1]

4 Dec 2023

PONE-D-22-29782R1Using simulations to compare the current Davis Cup ranking system to EloPLOS ONE

Dear Dr. Kelley,

Thank you for submitting your manuscript to PLOS ONE. After careful consideration, we feel that it has merit but does not fully meet PLOS ONE’s publication criteria as it currently stands. Therefore, we invite you to submit a revised version of the manuscript that addresses the points raised during the review process.

**ACADEMIC EDITOR: **Dear Authors,Thanks for submitting your work to PONE. Two referees read it and found some merits. However, both raised some concerns to be resolved carefully. Please go through the comments and address them point by point.

We look forward to receiving your revised manuscript.

Kind regards,

Erfan Babaee Tirkolaee, PhD

Academic Editor

PLOS ONE

Reviewers' comments:

Reviewer's Responses to Questions

**Comments to the Author**

1. If the authors have adequately addressed your comments raised in a previous round of review and you feel that this manuscript is now acceptable for publication, you may indicate that here to bypass the “Comments to the Author” section, enter your conflict of interest statement in the “Confidential to Editor” section, and submit your "Accept" recommendation.

Reviewer #3: (No Response)

Reviewer #4: (No Response)

2. Is the manuscript technically sound, and do the data support the conclusions?

Reviewer #3: (No Response)

Reviewer #4: Yes

3. Has the statistical analysis been performed appropriately and rigorously? 

Reviewer #3: (No Response)

Reviewer #4: Yes

4. Have the authors made all data underlying the findings in their manuscript fully available?

Reviewer #3: (No Response)

Reviewer #4: Yes

5. Is the manuscript presented in an intelligible fashion and written in standard English?

Reviewer #3: (No Response)

Reviewer #4: Yes

6. Review Comments to the Author

Reviewer #3: The authors could provide good work. However, there are some concerns to be resolved.

The abstract needs to be improved. The first sentence in the abstract, it is necessary for the authors to add a sentence to describe the problem or motivation to focus on this topic. The second sentence should provide the literature gap. In the third sentence, the authors should say what you are doing, and then provide the empirical findings. Finally, the significance of the finding should be offered.

The organization of the Introduction section is very unsatisfactory, and it is very messy and hard to read. Thus, this section needs rewriting in order to make it crisp and the main points of the research methodology should be mentioned clearly. This will help the readers to appreciate the novelty of the research.

English presentation of the paper should be modified so as to be more readable.

Improve the literature review. Add several pieces of research in 2019-2023. Moreover, the following references should be used:

A robust possibilistic programming framework for designing an organ transplant supply chain under uncertainty. Annals of Operations Research, 1-38.

Two-echelon electric vehicle routing problem with a developed moth-flame meta-heuristic algorithm. Operations Management Research, 1-22.

Efficient multi-objective meta-heuristic algorithms for energy-aware non-permutation flow-shop scheduling problem. Expert Systems with Applications, 213, 119077.

Designing a portfolio-based closed-loop supply chain network for dairy products with a financial approach: Accelerated Benders decomposition algorithm. Computers & Operations Research, 155, 106244.

Integration of blockchain-enabled closed-loop supply chain and robust product portfolio design. Computers & Industrial Engineering, 179, 109211.

Just-in-time scheduling in identical parallel machine sequence-dependent group scheduling problem. Journal of Industrial & Management Optimization.

Most of the methodological choices lack a clear motivation, and their impact on performance is not analyzed on the manuscript. On the whole, there is no clear indication of where the authors see the main innovation and value of the methodology described.

Discuss the pros and cons of the proposed methodology in more detail.

The author has poorly discussed the results of the paper. One would expect to find the previous empirical work enriching the discussions of the results, but unfortunately, that has not been done.

The discussion is relatively simple and insufficient. I recommend strengthening the comparison with previous research. Please compare the results in this study with those in previous studies. Discuss the study findings here. The discussion and conclusion are appropriately written and require no changes. The manuscript does not answer the following concerns: Why is it timeliness to explore such a study? What makes this study different from the previously published studies? Are there any similarly findings in line with the previously published studies? Are the findings different from prior academic studies that were conducted elsewhere, if any?

Reviewer #4: The paper presents a comparative approach to better ranking system of Davis Cup. Although, the application is new to tennis, there are no significant novel methods. In addition, the results and achievements are significant and can be implemented.

Also, the content is adequate, but it is not well-organized and the layout of the manuscript is incorrect. Please consider changing the layout of the manuscript by introducing the main points:

1. Introduction

2. Materials and methods

3. Results and discussion

4. Conclusions

You can also number the main and sub-headings for better tracking.

7. PLOS authors have the option to publish the peer review history of their article (what does this mean?). If published, this will include your full peer review and any attached files.

Reviewer #3: No

Reviewer #4: No

---

## [Author Response · Author response to Decision Letter 1]

18 Jan 2024

Response to Reviewers

Reviewer #3: The authors could provide good work. However, there are some concerns to be resolved.

The abstract needs to be improved. The first sentence in the abstract, it is necessary for the authors to add a sentence to describe the problem or motivation to focus on this topic. 

I thank the reviewer for their suggestions. I assume the reviewer means “first paragraph” when they state “first sentence”. The sentence “Therefore, it is essential that the ranking method performs well with respect to required performance criteria of the International Tennis Federation.” Has been added to the end of the first paragraph of the abstract to describe the motivation for the work. 

The second sentence should provide the literature gap. 

I thank the reviewer for their suggestions. I assume the reviewer means “second paragraph” when they state “second sentence”. The sentence “Such a comparison has not previously been published.” Has been added to the 2nd paragraph to highlight the literature gap.

In the third sentence, the authors should say what you are doing, and then provide the empirical findings.

I thank the reviewer for their suggestions. I assume the reviewer means “paragraph when they state “sentence”. The second paragraph describes what was done and the third paragraph describes the findings.

Finally, the significance of the finding should be offered.

I thank the reviewer for their suggestions. The final paragraph of the abstract has been changed to: “In conclusion, the current Davis Cup ranking method is performing at a standard that cannot be matched by typically optimised Elo but can be matched when an alternative optimisation method is used. Therefore, no evidence was found to suggest that the current Davis Cup ranking method could be improved upon by using Elo. However, alternative K-factor optimisation methods should be considered when applying Elo to a competition.”

The organization of the Introduction section is very unsatisfactory, and it is very messy and hard to read. Thus, this section needs rewriting in order to make it crisp and the main points of the research methodology should be mentioned clearly. This will help the readers to appreciate the novelty of the research.

I thank the reviewer for their suggestions. Numbered headings have been included to allow the reader to more easily follow the document which is of particular use in the introduction.

English presentation of the paper should be modified so as to be more readable.

I thank the reviewer for their suggestions. The English has been thoroughly checked.

Improve the literature review. Add several pieces of research in 2019-2023. Moreover, the following references should be used:

A robust possibilistic programming framework for designing an organ transplant supply chain under uncertainty. Annals of Operations Research, 1-38.

Two-echelon electric vehicle routing problem with a developed moth-flame meta-heuristic algorithm. Operations Management Research, 1-22.

Efficient multi-objective meta-heuristic algorithms for energy-aware non-permutation flow-shop scheduling problem. Expert Systems with Applications, 213, 119077.

Designing a portfolio-based closed-loop supply chain network for dairy products with a financial approach: Accelerated Benders decomposition algorithm. Computers & Operations Research, 155, 106244.

Integration of blockchain-enabled closed-loop supply chain and robust product portfolio design. Computers & Industrial Engineering, 179, 109211.

Just-in-time scheduling in identical parallel machine sequence-dependent group scheduling problem. Journal of Industrial & Management Optimization.

I thank the reviewer for the suggestions. The following sentence has been added:

“Simulations-based optimisation has been previously used in medical supply chain design [1]”.

It is difficult to include the remaining suggested references. The improved moth-flame optimisation used in the 2nd suggested reference and the many optimisations used in the 3rd suggested reference are not necessary for or relevant to the optimisation carried out in this work. Additionally, the time-saving findings are not relevant as my analysis was not time-limited. The 4th suggested reference uses a Benders decomposition, not required here as we have a simple single parameter problem. I cannot see how the implementation of blockchain technology is relevant to this work with regard to the 5th suggested reference and it is hard to compare the performance of nations to a portfolio of products. The optimisation methods and the speed and solution accuracy findings of the 6th suggested reference are also not relevant to this work. I notice the reviewer has focussed solely on references authored by Alireza Goli. There may be better fitting additional references the reviewer has knowledge of beyond the work of Goli.

Most of the methodological choices lack a clear motivation, and their impact on performance is not analyzed on the manuscript. 

Performance in terms of computational speed was not within the scope of this research. The optimisation described was to select the “best” k-factor for the Elo algorithm. The standard cost function for Elo is the prediction of results. this is described in section 3.2. The innovative alternative cost function is also described in section 3.2. 

On the whole, there is no clear indication of where the authors see the main innovation and value of the methodology described.

The main methodological innovation and value is the finding that an innovative alternative optimisation for the Elo k-factor produced an algorithm that out-performed Elo optimised in the standard way, as stated in the Results, discussed in the Discussion and highlighted in the conclusion with implications stated. The main findings of the paper are not methodological – see the aim in section 2.5. 

Discuss the pros and cons of the proposed methodology in more detail.

The limitations are covered in the Limitations section (6.2).

The author has poorly discussed the results of the paper. One would expect to find the previous empirical work enriching the discussions of the results, but unfortunately, that has not been done.

The reviewer is not clear as to the section they are referring to. Do they mean the results section or the discussion section? This approach has not been applied before, either to compare different ranking methods or specifically to test Elo, so there is little empirical work to compare against

The discussion is relatively simple and insufficient. I recommend strengthening the comparison with previous research. Please compare the results in this study with those in previous studies. Discuss the study findings here. The discussion and conclusion are appropriately written and require no changes.

This reviewer comment is quite contradictory. I will assume final sentence is what is required.

The manuscript does not answer the following concerns: Why is it timeliness to explore such a study? What makes this study different from the previously published studies? Are there any similarly findings in line with the previously published studies? Are the findings different from prior academic studies that were conducted elsewhere, if any?

In response to this reviewer’s comment, I will quote reviewer 1’s comments: 

“The reviewer would also like to congratulate the author for having chosen as the subject of his study a considerably relevant aspect within the discipline of tennis, namely the analysis of ranking systems.

In fact, the issue of ranking systems for both individuals and teams participating in a competition is a very topical issue in the world of tennis due to the emergence of the ITF World Tennis Number.

Thus, the reviewer considers that we are faced with a tremendously current, modern and necessary study due to its obvious implications both for the participating federations and for the players who are members of the national teams.

The author adequately justifies the importance of the topic, explains its necessity and introduces it in an appropriate manner in the opening section of the article.”

Reviewer #4: The paper presents a comparative approach to better ranking system of Davis Cup. Although, the application is new to tennis, there are no significant novel methods. In addition, the results and achievements are significant and can be implemented.

Also, the content is adequate, but it is not well-organized and the layout of the manuscript is incorrect. Please consider changing the layout of the manuscript by introducing the main points:

1. Introduction

2. Materials and methods

3. Results and discussion

4. Conclusions

You can also number the main and sub-headings for better tracking.

I thank the reviewer for their comments. I have reformatted and restructured the headings to fit the proposed headings above and included numbered headings and sub-headings as suggested. The results and discussion section have been kept separate, but the sub-headings of the discussion match the relevant sub-headings in the results.

---

## [Decision Letter · Decision Letter 2]

22 Jan 2024

Using simulations to compare the current Davis Cup ranking system to Elo

PONE-D-22-29782R2

Dear Dr. Kelley,

We’re pleased to inform you that your manuscript has been judged scientifically suitable for publication and will be formally accepted for publication once it meets all outstanding technical requirements.

Kind regards,

Erfan Babaee Tirkolaee, PhD

Academic Editor

PLOS ONE

Additional Editor Comments (optional):

Reviewers' comments:

Reviewer's Responses to Questions

**Comments to the Author**

1. If the authors have adequately addressed your comments raised in a previous round of review and you feel that this manuscript is now acceptable for publication, you may indicate that here to bypass the “Comments to the Author” section, enter your conflict of interest statement in the “Confidential to Editor” section, and submit your "Accept" recommendation.

Reviewer #3: All comments have been addressed

Reviewer #4: All comments have been addressed

2. Is the manuscript technically sound, and do the data support the conclusions?

Reviewer #3: Yes

Reviewer #4: Yes

3. Has the statistical analysis been performed appropriately and rigorously? 

Reviewer #3: Yes

Reviewer #4: Yes

4. Have the authors made all data underlying the findings in their manuscript fully available?

Reviewer #3: Yes

Reviewer #4: Yes

5. Is the manuscript presented in an intelligible fashion and written in standard English?

Reviewer #3: Yes

Reviewer #4: Yes

6. Review Comments to the Author

Reviewer #3: Authors have improved the paper well and it can be published.

Reviewer #4: (No Response)

7. PLOS authors have the option to publish the peer review history of their article (what does this mean?). If published, this will include your full peer review and any attached files.

Reviewer #3: No

Reviewer #4: No

---

## [Editor Report · Acceptance letter]

16 Feb 2024

PONE-D-22-29782R2 

PLOS ONE

Dear Dr. Kelley, 

I'm pleased to inform you that your manuscript has been deemed suitable for publication in PLOS ONE. Congratulations! Your manuscript is now being handed over to our production team.

Kind regards, 

on behalf of

Dr. Erfan Babaee Tirkolaee 

Academic Editor

PLOS ONE